# Submodular Optimization with Submodular Cover and Submodular Knapsack Constraints

**Rishabh Iyer**
Department of Electrical Engineering
University of Washington
rkiyer@u.washington.edu

**Jeff Bilmes**
Department of Electrical Engineering
University of Washington
bilmes@u.washington.edu

## Abstract

We investigate two new optimization problems — minimizing a submodular function subject to a submodular lower bound constraint (submodular cover) and maximizing a submodular function subject to a submodular upper bound constraint (submodular knapsack). We are motivated by a number of real-world applications in machine learning including sensor placement and data subset selection, which require maximizing a certain submodular function (like coverage or diversity) while simultaneously minimizing another (like cooperative cost). These problems are often posed as minimizing the difference between submodular functions [9, 25] which is in the worst case inapproximable. We show, however, that by phrasing these problems as constrained optimization, which is more natural for many applications, we achieve a number of bounded approximation guarantees. We also show that both these problems are closely related and an approximation algorithm solving one can be used to obtain an approximation guarantee for the other. We provide hardness results for both problems thus showing that our approximation factors are tight up to $\log$-factors. Finally, we empirically demonstrate the performance and good scalability properties of our algorithms.

## 1 Introduction

A set function $f : 2^V \to \mathbb{R}$ is said to be *submodular* [4] if for all subsets $S, T \subseteq V$, it holds that $f(S) + f(T) \geq f(S \cup T) + f(S \cap T)$. Defining $f(j|S) \triangleq f(S \cup j) - f(S)$ as the gain of $j \in V$ in the context of $S \subseteq V$, then $f$ is submodular if and only if $f(j|S) \geq f(j|T)$ for all $S \subseteq T$ and $j \notin T$. The function $f$ is monotone iff $f(j|S) \geq 0, \forall j \notin S, S \subseteq V$. For convenience, we assume the ground set is $V = \{1, 2, \cdots, n\}$. While general set function optimization is often intractable, many forms of submodular function optimization can be solved near optimally or even optimally in certain cases. Submodularity, moreover, is inherent in a large class of real-world applications, particularly in machine learning, therefore making them extremely useful in practice.

In this paper, we study a new class of discrete optimization problems that have the following form:

Problem 1 (SCSC): $\min\{f(X) \,|\, g(X) \geq c\}$, and Problem 2 (SCSK): $\max\{g(X) \,|\, f(X) \leq b\}$,

where $f$ and $g$ are monotone non-decreasing submodular functions that also, w.l.o.g., are normalized $(f(\emptyset) = g(\emptyset) = 0)$[1], and where $b$ and $c$ refer to budget and cover parameters respectively. The corresponding constraints are called the submodular cover [29] and submodular knapsack [1] respectively and hence we refer to Problem 1 as *Submodular Cost Submodular Cover* (henceforth SCSC) and Problem 2 as *Submodular Cost Submodular Knapsack* (henceforth SCSK). Our motivation stems from an interesting class of problems that require minimizing a certain submodular function $f$ while simultaneously maximizing another submodular function $g$. We shall see that these naturally

occur in applications like sensor placement, data subset selection, and many other machine learning applications. A standard approach used in literature [9, 25, 15] has been to transform these problems into minimizing the difference between submodular functions (also called DS optimization):

$$\text{Problem 0:} \quad \min_{X \subseteq V} \big( f(X) - g(X) \big). \tag{1}$$

While a number of heuristics are available for solving Problem 0, in the worst-case it is NP-hard and inapproximable [9], even when $f$ and $g$ are monotone. Although an exact branch and bound algorithm has been provided for this problem [15], its complexity can be exponential in the worst case. On the other hand, in many applications, one of the submodular functions naturally serves as part of a constraint. For example, we might have a budget on a cooperative cost, in which case Problems 1 and 2 become applicable. The utility of Problems 1 and 2 become apparent when we consider how they occur in real-world applications and how they subsume a number of important optimization problems.

**Sensor Placement and Feature Selection:** Often, the problem of choosing sensor locations can be modeled [19, 9] by maximizing the mutual information between the chosen variables $A$ and the unchosen set $V \setminus A$ (i.e., $f(A) = I(X_A; X_{V \setminus A})$). Alternatively, we may wish to maximize the mutual information between a set of chosen sensors $X_A$ and a quantity of interest $C$ (i.e., $f(A) = I(X_A; C)$) assuming that the set of features $X_A$ are conditionally independent given $C$ [19, 9]. Both these functions are submodular. Since there are costs involved, we want to simultaneously minimize the cost $g(A)$. Often this cost is submodular [19, 9]. For example, there is typically a discount when purchasing sensors in bulk (economies of scale). This then becomes a form of either Problem 1 or 2.

**Data subset selection:** A data subset selection problem in speech and NLP involves finding a limited vocabulary which simultaneously has a large coverage. This is particularly useful, for example in speech recognition and machine translation, where the complexity of the algorithm is determined by the vocabulary size. The motivation for this problem is to find the subset of training examples which will facilitate evaluation of prototype systems [23]. Often the objective functions encouraging small vocabulary subsets and large acoustic spans are submodular [23, 20] and hence this problem can naturally be cast as an instance of Problems 1 and 2.

**Privacy Preserving Communication:** Given a set of random variables $X_1, \cdots, X_n$, denote $\mathfrak{I}$ as an information source, and $\mathfrak{P}$ as private information that should be filtered out. Then one way of formulating the problem of choosing a information containing but privacy preserving set of random variables can be posed as instances of Problems 1 and 2, with $f(A) = H(X_A | \mathfrak{I})$ and $g(A) = H(X_A | \mathfrak{P})$, where $H(\cdot | \cdot)$ is the conditional entropy.

**Machine Translation**: Another application in machine translation is to choose a subset of training data that is optimized for given test data set, a problem previously addressed with modular functions [24]. Defining a submodular function with ground set over the union of training and test sample inputs $V = V_{\text{tr}} \cup V_{\text{te}}$, we can set $f : 2^{V_{\text{tr}}} \to \mathbb{R}_+$ to $f(X) = f(X | V_{\text{te}})$, and take $g(X) = |X|$, and $b \approx 0$ in Problem 2 to address this problem. We call this the *Submodular Span* problem.

Apart from the real-world applications above, both Problems 1 and 2 generalize a number of well-studied discrete optimization problems. For example the *Submodular Set Cover* problem (henceforth SSC) [29] occurs as a special case of Problem 1, with $f$ being modular and $g$ is submodular. Similarly the *Submodular Cost Knapsack* problem (henceforth SK) [28] is a special case of problem 2 again when $f$ is modular and $g$ submodular. Both these problems subsume the *Set Cover* and *Max k-Cover* problems [3]. When both $f$ and $g$ are modular, Problems 1 and 2 are called *knapsack problems* [16].

The following are some of our contributions. We show that Problems 1 and 2 are intimately connected, in that any approximation algorithm for either problem can be used to provide guarantees for the other problem as well. We then provide a framework of combinatorial algorithms based on optimizing, sometimes iteratively, subproblems that are easy to solve. These subproblems are obtained by computing either upper or lower bound approximations of the cost functions or constraining functions. We also show that many combinatorial algorithms like the greedy algorithm for SK [28] and SSC [29] also belong to this framework and provide the first constant-factor bi-criterion approximation algorithm for SSC [29] and hence the general set cover problem [3]. We then show how with suitable choices of approximate functions, we can obtain a number of bounded approximation guarantees and show the hardness for Problems 1 and 2, which in fact match some of our approximation guarantees. Our guarantees and hardness results depend on the *curvature* of the submodular functions [2]. We observe a strong asymmetry in the results that the factors change

polynomially based on the curvature of $f$ but only by a constant-factor with the curvature of $g$, hence making the SK and SSC much easier compared to SCSK and SCSC.

## 2  Background and Main Ideas

We first introduce several key concepts used throughout the paper. This paper includes only the main results and we defer all the proofs and additional discussions to the extended version [11]. Given a submodular function $f$, we define the total curvature, $\kappa_f$ as[2]: $\kappa_f = 1 - \min_{j \in V} \frac{f(j|V\setminus j)}{f(j)}$ [2]. Intuitively, the curvature $0 \leq \kappa_f \leq 1$ measures the distance of $f$ from modularity and $\kappa_f = 0$ if and only if $f$ is modular (or additive, i.e., $f(X) = \sum_{j \in X} f(j)$). A number of approximation guarantees in the context of submodular optimization have been refined via the curvature of the submodular function [2, 13, 12]. In this paper, we shall witness the role of curvature also in determining the approximations and the hardness of problems 1 and 2.

The main idea of this paper is a framework of algorithms based on choosing appropriate surrogate functions for $f$ and $g$ to optimize over. This framework is represented in Algorithm 1. We would like to choose surrogate functions $\hat{f}_t$ and $\hat{g}_t$ such that using them, Problems 1 and 2 become easier. If the algorithm is just single stage (not iterative), we represent the surrogates as $\hat{f}$ and $\hat{g}$. The surrogate functions we consider in this paper are in the forms of bounds (upper or lower) and approximations.

---

**Algorithm 1:** General algorithmic framework to address both Problems 1 and 2

---
1: **for** $t = 1, 2, \cdots, T$ **do**
2:     Choose surrogate functions $\hat{f}_t$ and $\hat{g}_t$ for $f$ and $g$ respectively, tight at $X^{t-1}$.
3:     Obtain $X^t$ as the optimizer of Problem 1 or 2 with $\hat{f}_t$ and $\hat{g}_t$ instead of $f$ and $g$.
4: **end for**

---

**Modular lower bounds:** Akin to convex functions, submodular functions have tight modular lower bounds. These bounds are related to the subdifferential $\partial_f(Y)$ of the submodular set function $f$ at a set $Y \subseteq V$ [4]. Denote a subgradient at $Y$ by $h_Y \in \partial_f(Y)$. The extreme points of $\partial_f(Y)$ may be computed via a greedy algorithm: Let $\pi$ be a permutation of $V$ that assigns the elements in $Y$ to the first $|Y|$ positions ($\pi(i) \in Y$ if and only if $i \leq |Y|$). Each such permutation defines a chain with elements $S_0^\pi = \emptyset$, $S_i^\pi = \{\pi(1), \pi(2), \ldots, \pi(i)\}$ and $S_{|Y|}^\pi = Y$. This chain defines an extreme point $h_Y^\pi$ of $\partial_f(Y)$ with entries $h_Y^\pi(\pi(i)) = f(S_i^\pi) - f(S_{i-1}^\pi)$. Defined as above, $h_Y^\pi$ forms a lower bound of $f$, tight at $Y$ — i.e., $h_Y^\pi(X) = \sum_{j \in X} h_Y^\pi(j) \leq f(X), \forall X \subseteq V$ and $h_Y^\pi(Y) = f(Y)$.

**Modular upper bounds:** We can also define superdifferentials $\partial^f(Y)$ of a submodular function [14, 10] at $Y$. It is possible, moreover, to provide specific supergradients [10, 13] that define the following two modular upper bounds (when referring either one, we use $m_X^f$):

$$m_{X,1}^f(Y) \triangleq f(X) - \sum_{j \in X\setminus Y} f(j|X\setminus j) + \sum_{j \in Y\setminus X} f(j|\emptyset), \quad m_{X,2}^f(Y) \triangleq f(X) - \sum_{j \in X\setminus Y} f(j|V\setminus j) + \sum_{j \in Y\setminus X} f(j|X).$$

Then $m_{X,1}^f(Y) \geq f(Y)$ and $m_{X,2}^f(Y) \geq f(Y), \forall Y \subseteq V$ and $m_{X,1}^f(X) = m_{X,2}^f(X) = f(X)$.

**MM algorithms using upper/lower bounds:** Using the modular upper and lower bounds above in Algorithm 1, provide a class of Majorization-Minimization (MM) algorithms, akin to the algorithms proposed in [13] for submodular optimization and in [25, 9] for DS optimization (Problem 0 above). An appropriate choice of the bounds ensures that the algorithm always improves the objective values for Problems 1 and 2. In particular, choosing $\hat{f}_t$ as a modular upper bound of $f$ tight at $X^t$, or $\hat{g}_t$ as a modular lower bound of $g$ tight at $X^t$, or both, ensures that the objective value of Problems 1 and 2 always improves at every iteration as long as the corresponding surrogate problem can be solved exactly. Unfortunately, Problems 1 and 2 are NP-hard even if $f$ or $g$ (or both) are modular [3], and therefore the surrogate problems themselves cannot be solved exactly. Fortunately, the surrogate problems are often much easier than the original ones and can admit $\log$ or constant-factor guarantees. In practice, moreover, these factors are almost 1. Furthermore, with a simple modification of the iterative procedure of Algorithm 1, we can guarantee improvement at every iteration [11]. What is also fortunate and perhaps surprising, as we show in this paper below, is that unlike the case of DS optimization (where the problem is inapproximable in general [9]), the constrained forms of optimization (Problems 1 and 2) do have approximation guarantees.

**Ellipsoidal Approximation:** We also consider ellipsoidal approximations (EA) of $f$. The main result of Goemans et. al [6] is to provide an algorithm based on approximating the submodular polyhedron by an ellipsoid. They show that for any polymatroid function $f$, one can compute an approximation of the form $\sqrt{w^f(X)}$ for a certain modular weight vector $w^f \in \mathbb{R}^V$, such that $\sqrt{w^f(X)} \leq f(X) \leq O(\sqrt{n}\log n)\sqrt{w^f(X)}, \forall X \subseteq V$. A simple trick then provides a curvature-dependent approximation [12] — we define the $\kappa_f$-*curve-normalized* version of $f$ as follows: $f^\kappa(X) \triangleq \left[f(X) - (1 - \kappa_f)\sum_{j\in X} f(j)\right]/\kappa_f$. Then, the submodular function $f^{\text{ea}}(X) = \kappa_f\sqrt{w^{f^\kappa}(X)} + (1 - \kappa_f)\sum_{j\in X} f(j)$ satisfies [12]:

$$f^{\text{ea}}(X) \leq f(X) \leq O\left(\frac{\sqrt{n}\log n}{1 + (\sqrt{n}\log n - 1)(1 - \kappa_f)}\right) f^{\text{ea}}(X), \forall X \subseteq V \tag{2}$$

$f^{\text{ea}}$ is multiplicatively bounded by $f$ by a factor depending on $\sqrt{n}$ and the curvature. We shall use the result above in providing approximation bounds for Problems 1 and 2. In particular, the surrogate functions $\hat{f}$ or $\hat{g}$ in Algorithm 1 can be the ellipsoidal approximations above, and the multiplicative bounds transform into approximation guarantees for these problems.

## 3   Relation between SCSC and SCSK

In this section, we show a precise relationship between Problems 1 and 2. From the formulation of Problems 1 and 2, it is clear that these problems are duals of each other. Indeed, in this section we show that the problems are polynomially transformable into each other.

| **Algorithm 2:** Approx. algorithm for SCSK using an approximation algorithm for SCSC. | **Algorithm 3:** Approx. algorithm for SCSC using an approximation algorithm for SCSK. |
|---|---|
| 1: **Input:** An SCSK instance with budget $b$, an $[\sigma, \rho]$ approx. algo. for SCSC, & $\epsilon \in [0, 1)$. | 1: **Input:** An SCSC instance with cover $c$, an $[\rho, \sigma]$ approx. algo. for SCSK, & $\epsilon > 0$. |
| 2: **Output:** $[(1 - \epsilon)\rho, \sigma]$ approx. for SCSK. | 2: **Output:** $[(1 + \epsilon)\sigma, \rho]$ approx. for SCSC. |
| 3: $c \leftarrow g(V), \hat{X}_c \leftarrow V$. | 3: $b \leftarrow \mathrm{argmin}_j f(j), \hat{X}_b \leftarrow \emptyset$. |
| 4: **while** $f(\hat{X}_c) > \sigma b$ **do** | 4: **while** $g(\hat{X}_b) < \rho c$ **do** |
| 5:    $c \leftarrow (1 - \epsilon)c$ | 5:    $b \leftarrow (1 + \epsilon)b$ |
| 6:    $\hat{X}_c \leftarrow [\sigma, \rho]$ approx. for SCSC using $c$. | 6:    $\hat{X}_b \leftarrow [\rho, \sigma]$ approx. for SCSK using $b$. |
| 7: **end while** | 7: **end while** |

We first introduce the notion of bicriteria algorithms. An algorithm is a $[\sigma, \rho]$ bi-criterion algorithm for Problem 1 if it is guaranteed to obtain a set $X$ such that $f(X) \leq \sigma f(X^*)$ (approximate optimality) and $g(X) \geq c' = \rho c$ (approximate feasibility), where $X^*$ is an optimizer of Problem 1. Similarly, an algorithm is a $[\rho, \sigma]$ bi-criterion algorithm for Problem 2 if it is guaranteed to obtain a set $X$ such that $g(X) \geq \rho g(X^*)$ and $f(X) \leq b' = \sigma b$, where $X^*$ is the optimizer of Problem 2. In a bi-criterion algorithm for Problems 1 and 2, typically $\sigma \geq 1$ and $\rho \leq 1$. A *non-bicriterion* algorithm for Problem 1 is when $\rho = 1$ and a *non-bicriterion* algorithm for Problem 2 is when $\sigma = 1$. Algorithms 2 and 3 provide the schematics for using an approximation algorithm for one of the problems for solving the other.

**Theorem 3.1.** *Algorithm 2 is guaranteed to find a set $\hat{X}_c$ which is a $[(1 - \epsilon)\rho, \sigma]$ approximation of SCSK in at most $\log_{1/(1-\epsilon)}[g(V)/\min_j g(j)]$ calls to the $[\sigma, \rho]$ approximate algorithm for SCSC. Similarly, Algorithm 3 is guaranteed to find a set $\hat{X}_b$ which is a $[(1 + \epsilon)\sigma, \rho]$ approximation of SCSC in $\log_{1+\epsilon}[f(V)/\min_j f(j)]$ calls to a $[\rho, \sigma]$ approximate algorithm for SCSK.*

Theorem 3.1 implies that the complexity of Problems 1 and 2 are identical, and a solution to one of them provides a solution to the other. Furthermore, as expected, the hardness of Problems 1 and 2 are also almost identical. When $f$ and $g$ are polymatroid functions, moreover, we can provide bounded approximation guarantees for both problems, as shown in the next section. Alternatively we can also do a binary search instead of a linear search to transform Problems 1 and 2. This essentially turns the factor of $O(1/\epsilon)$ into $O(\log 1/\epsilon)$. Due to lack of space, we defer this discussion to the extended version [11].

# 4 Approximation Algorithms

We consider several algorithms for Problems 1 and 2, which can all be characterized by the framework of Algorithm 1, using the surrogate functions of the form of upper/lower bounds or approximations.

## 4.1 Approximation Algorithms for SCSC

We first describe our approximation algorithms designed specifically for SCSC, leaving to §4.2 the presentation of our algorithms slated for SCSK. We first investigate a special case, the submodular set cover (SSC), and then provide two algorithms, one of them (ISSC) is very practical with a weaker theoretical guarantee, and another one (EASSC) which is slow but has the tightest guarantee.

**Submodular Set Cover (SSC):** We start by considering a classical special case of SCSC (Problem 1) where $f$ is already a modular function and $g$ is a submodular function. This problem occurs naturally in a number of problems related to active/online learning [7] and summarization [21, 22]. This problem was first investigated by Wolsey [29], wherein he showed that a simple greedy algorithm achieves bounded (in fact, log-factor) approximation guarantees. We show that this greedy algorithm can naturally be viewed in the framework of our Algorithm 1 by choosing appropriate surrogate functions $\hat{f}_t$ and $\hat{g}_t$. The idea is to use the modular function $f$ as its own surrogate $\hat{f}_t$ and choose the function $\hat{g}_t$ as a modular lower bound of $g$. Akin to the framework of algorithms in [13], the crucial factor is the choice of the lower bound (or subgradient). Define the *greedy subgradient* as:

$$\pi(i) \in \operatorname{argmin} \left\{ \frac{f(j)}{g(j|S_{i-1}^{\pi})} \; \middle| \; j \notin S_{i-1}^{\pi}, g(S_{i-1}^{\pi} \cup j) < c \right\}. \tag{3}$$

Once we reach an $i$ where the constraint $g(S_{i-1}^{\pi} \cup j) < c$ can no longer be satisfied by any $j \notin S_{i-1}^{\pi}$, we choose the remaining elements for $\pi$ arbitrarily. Let the corresponding subgradient be referred to as $h^{\pi}$. Then we have the following lemma, which is an extension of [29], and which is a simpler description of the result stated formally in [11].

**Lemma 4.1.** *The greedy algorithm for SSC [29] can be seen as an instance of Algorithm 1 by choosing the surrogate function $\hat{f}$ as $f$ and $\hat{g}$ as $h^{\pi}$ (with $\pi$ defined in Eqn. (3)).*

When $g$ is integral, the guarantee of the greedy algorithm is $H_g \triangleq H(\max_j g(j))$, where $H(d) = \sum_{i=1}^{d} \frac{1}{i}$ [29] (henceforth we will use $H_g$ for this quantity). This factor is tight up to lower-order terms [3]. Furthermore, since this algorithm directly solves SSC, we call it the *primal greedy*. We could also solve SSC by looking at its *dual*, which is SK [28]. Although SSC does not admit any constant-factor approximation algorithms [3], we can obtain a constant-factor *bi-criterion* guarantee:

**Lemma 4.2.** *Using the greedy algorithm for SK [28] as the approximation oracle in Algorithm 3 provides a $[1 + \epsilon, 1 - e^{-1}]$ bi-criterion approximation algorithm for SSC, for any $\epsilon > 0$.*

We call this the *dual greedy*. This result follows immediately from the guarantee of the submodular cost knapsack problem [28] and Theorem 3.1. We remark that we can also use a simpler version of the greedy iteration at every iteration [21, 17] and we obtain a guarantee of $(1 + \epsilon, 1/2(1 - e^{-1}))$. In practice, however, both these factors are almost 1 and hence the simple variant of the greedy algorithm suffices.

**Iterated Submodular Set Cover (ISSC):** We next investigate an algorithm for the general SCSC problem when both $f$ and $g$ are submodular. The idea here is to iteratively solve the submodular set cover problem which can be done by replacing $f$ by a modular upper bound at every iteration. In particular, this can be seen as a variant of Algorithm 1, where we start with $X^0 = \emptyset$ and choose $\hat{f}_t(X) = m_{X^t}^f(X)$ at every iteration. The surrogate problem at each iteration becomes $\min\{m_{X^t}^f(X)|g(X) \geq c\}$. Hence, each iteration is an instance of SSC and can be solved nearly optimally using the greedy algorithm. We can continue this algorithm for $T$ iterations or until convergence. An analysis very similar to the ones in [9, 13] will reveal polynomial time convergence. Since each iteration is only the greedy algorithm, this approach is also highly practical and scalable.

**Theorem 4.3.** *ISSC obtains an approximation factor of $\frac{K_g H_g}{1+(K_g-1)(1-\kappa_f)} \leq \frac{n}{1+(n-1)(1-\kappa_f)} H_g$ where $K_g = 1 + \max\{|X| : g(X) < c\}$ and $H_g$ is the approximation factor of the submodular set cover using g.*

From the above, it is clear that $K_g \leq n$. Notice also that $H_g$ is essentially a log-factor. We also see an interesting effect of the curvature $\kappa_f$ of $f$. When $f$ is modular ($\kappa_f = 0$), we recover the approximation guarantee of the submodular set cover problem. Similarly, when $f$ has restricted curvature, the guarantees can be much better. Moreover, the approximation guarantee already holds after the first iteration, so additional iterations can only further improve the objective.

**Ellipsoidal Approximation based Submodular Set Cover (EASSC):** In this setting, we use the ellipsoidal approximation discussed in §2. We can compute the $\kappa_f$-curve-normalized version of $f$ ($f^\kappa$, see §2), and then compute its ellipsoidal approximation $\sqrt{w^{f^\kappa}}$. We then define the function $\hat{f}(X) = f^{ea}(X) = \kappa_f \sqrt{w^{f^\kappa}(X)} + (1 - \kappa_f) \sum_{j \in X} f(j)$ and use this as the surrogate function $\hat{f}$ for $f$. We choose $\hat{g}$ as $g$ itself. The surrogate problem becomes:

$$\min \left\{ \kappa_f \sqrt{w^{f^\kappa}(X)} + (1 - \kappa_f) \sum_{j \in X} f(j) \, \middle| \, g(X) \geq c \right\}. \tag{4}$$

While function $\hat{f}(X) = f^{ea}(X)$ is not modular, it is a weighted sum of a concave over modular function and a modular function. Fortunately, we can use the result from [26], where they show that any function of the form of $\sqrt{w_1(X)} + w_2(X)$ can be optimized over any polytope $\mathcal{P}$ with an approximation factor of $\beta(1 + \epsilon)$ for any $\epsilon > 0$, where $\beta$ is the approximation factor of optimizing a modular function over $\mathcal{P}$. The complexity of this algorithm is polynomial in $n$ and $\frac{1}{\epsilon}$. We use their algorithm to minimize $f^{ea}(X)$ over the *submodular set cover* constraint and hence we call this algorithm EASSC.

**Theorem 4.4.** *EASSC obtains a guarantee of $O(\frac{\sqrt{n} \log n H_g}{1 + (\sqrt{n} \log n - 1)(1 - \kappa_f)})$, where $H_g$ is the approximation guarantee of the set cover problem.*

If the function $f$ has $\kappa_f = 1$, we can use a much simpler algorithm. In particular, we can minimize $(f^{ea}(X))^2 = w^f(X)$ at every iteration, giving a surrogate problem of the form $\min\{w^f(X)|g(X) \geq c\}$. This is directly an instance of SSC, and in contrast to EASSC, we just need to solve SSC once. We call this algorithm EASSCc.

**Corollary 4.5.** *EASSCc obtains an approximation guarantee of $O(\sqrt{n} \log n \sqrt{H_g})$.*

## 4.2 Approximation Algorithms for SCSK

In this section, we describe our approximation algorithms for SCSK. We note the dual nature of the algorithms in this current section to those given in §4.1. We first investigate a special case, the submodular knapsack (SK), and then provide three algorithms, two of them (Gr and ISK) being practical with slightly weaker theoretical guarantee, and another one (EASK) which is not scalable but has the tightest guarantee.

**Submodular Cost Knapsack (SK):** We start with a special case of SCSK (Problem 2), where $f$ is a modular function and $g$ is a submodular function. In this case, SCSK turns into the SK problem for which the greedy algorithm with partial enumeration provides a $1 - e^{-1}$ approximation [28]. The greedy algorithm can be seen as an instance of Algorithm 1 with $\hat{g}$ being the modular lower bound of $g$ and $\hat{f}$ being $f$, which is already modular. In particular, define:

$$\pi(i) \in \operatorname{argmax} \left\{ \frac{g(j|S_{i-1}^\pi)}{f(j)} \, \middle| \, j \notin S_{i-1}^\pi, f(S_{i-1}^\pi \cup \{j\}) \leq b \right\}, \tag{5}$$

where the remaining elements are chosen arbitrarily. The following is an informal description of the result described formally in [11].

**Lemma 4.6.** *Choosing the surrogate function $\hat{f}$ as $f$ and $\hat{g}$ as $h^\pi$ (with $\pi$ defined in eqn (5)) in Algorithm 1 with appropriate initialization obtains a guarantee of $1 - 1/e$ for SK.*

**Greedy (Gr):** A similar greedy algorithm can provide approximation guarantees for the general SCSK problem, with submodular $f$ and $g$. Unlike the knapsack case in (5), however, at iteration $i$ we choose an element $j \notin S_{i-1} : f(S_{i-1}^\pi \cup \{j\}) \leq b$ which maximizes $g(j|S_{i-1})$. In terms of Algorithm 1, this is analogous to choosing a permutation, $\pi$ such that:

$$\pi(i) \in \operatorname{argmax}\{g(j|S_{i-1}^\pi)|j \notin S_{i-1}^\pi, f(S_{i-1}^\pi \cup \{j\}) \leq b\}. \tag{6}$$

**Theorem 4.7.** *The greedy algorithm for SCSK obtains an approx. factor of* $\frac{1}{\kappa_g}(1 - (\frac{K_f - \kappa_g}{K_f})^{k_f}) \geq \frac{1}{K_f}$, *where* $K_f = \max\{|X| : f(X) \leq b\}$ *and* $k_f = \min\{|X| : f(X) \leq b \, \& \, \forall j \in X, f(X \cup j) > b\}$.

In the worst case, $k_f = 1$ and $K_f = n$, in which case the guarantee is $1/n$. The bound above follows from a simple observation that the constraint $\{f(X) \leq b\}$ is down-monotone for a monotone function $f$. However, in this variant, we do not use any specific information about $f$. In particular it holds for maximizing a submodular function $g$ over any down monotone constraint [2]. Hence it is conceivable that an algorithm that uses both $f$ and $g$ to choose the next element could provide better bounds. We do not, however, currently have the analysis for this.

**Iterated Submodular Cost Knapsack (ISK):** Here, we choose $\hat{f}_t(X)$ as a modular upper bound of $f$, tight at $X^t$. Let $\hat{g}_t = g$. Then at every iteration, we solve $\max\{g(X) | m^f_{X^t}(X) \leq b\}$, which is a submodular maximization problem subject to a knapsack constraint (SK). As mentioned above, greedy can solve this nearly optimally. We start with $X^0 = \emptyset$, choose $\hat{f}_0(X) = \sum_{j \in X} f(j)$ and then iteratively continue this process until convergence (note that this is an ascent algorithm). We have the following theoretical guarantee:

**Theorem 4.8.** *Algorithm ISK obtains a set* $X^t$ *such that* $g(X^t) \geq (1-e^{-1})g(\tilde{X})$, *where* $\tilde{X}$ *is the optimal solution of* $\max\left\{g(X) \mid f(X) \leq \frac{b(1+(K_f-1)(1-\kappa_f))}{K_f}\right\}$ *and where* $K_f = \max\{|X| : f(X) \leq b\}$.

It is worth pointing out that the above bound holds even after the first iteration of the algorithm. It is interesting to note the similarity between this approach and ISSC. Notice that the guarantee above is not a standard bi-criterion approximation. We show in the extended version [11] that with a simple transformation, we can obtain a bicriterion guarantee.

**Ellipsoidal Approximation based Submodular Cost Knapsack (EASK):** Choosing the Ellipsoidal Approximation $f^{ea}$ of $f$ as a surrogate function, we obtain a simpler problem:

$$\max\left\{g(X) \,\middle|\, \kappa_f\sqrt{w^{f^\kappa}(X)} + (1-\kappa_f)\sum_{j \in X} f(j) \leq b\right\}. \tag{7}$$

In order to solve this problem, we look at its dual problem (i.e., Eqn. (4)) and use Algorithm 2 to convert the guarantees. We call this procedure EASK. We then obtain guarantees very similar to Theorem 4.4.

**Lemma 4.9.** *EASK obtains a guarantee of* $\left[1 + \epsilon, O(\frac{\sqrt{n}\log n H_g}{1+(\sqrt{n}\log n-1)(1-\kappa_f)})\right]$.

In the case when the submodular function has a curvature $\kappa_f = 1$, we can actually provide a simpler algorithm without needing to use the conversion algorithm (Algorithm 2). In this case, we can directly choose the ellipsoidal approximation of $f$ as $\sqrt{w^f(X)}$ and solve the surrogate problem: $\max\{g(X) : w^f(X) \leq b^2\}$. This surrogate problem is a submodular cost knapsack problem, which we can solve using the greedy algorithm. We call this algorithm EASKc. This guarantee is tight up to log factors if $\kappa_f = 1$.

**Corollary 4.10.** *Algorithm EASKc obtains a bi-criterion guarantee of* $[1 - e^{-1}, O(\sqrt{n}\log n)]$.

### 4.3 Extensions beyond SCSC and SCSK

SCSC and SCSK can in fact be extended to more flexible and complicated constraints which can arise naturally in many applications [18, 8]. These include multiple covering and knapsack constraints – i.e., $\min\{f(X)|g_i(X) \geq c_i, i = 1, 2, \cdots k\}$ and $\max\{g(X)|f_i(X) \leq b_i, i = 1, 2, \cdots k\}$, and robust optimization problems like $\max\{\min_i g_i(X)|f(X) \leq b\}$, where the functions $f, g, f_i$'s and $g_i$'s are submodular. We also consider SCSC and SCSK with non-monotone submodular functions. Due to lack of space, we defer these discussions to the extended version of this paper [11].

### 4.4 Hardness

In this section, we provide the hardness for Problems 1 and 2. The lower bounds serve to show that the approximation factors above are almost tight.

**Theorem 4.11.** *For any $\kappa > 0$, there exists submodular functions with curvature $\kappa$ such that no polynomial time algorithm for Problems 1 and 2 achieves a bi-criterion factor better than $\frac{\sigma}{\rho} = \frac{n^{1/2-\epsilon}}{1+(n^{1/2-\epsilon}-1)(1-\kappa)}$ for any $\epsilon > 0$.*

The above result shows that EASSC and EASK meet the bounds above to log factors. We see an interesting curvature-dependent influence on the hardness. We also see this phenomenon in the approximation guarantees of our algorithms. In particular, as soon as $f$ becomes modular, the problem becomes easy, even when $g$ is submodular. This is not surprising since the submodular set cover problem and the submodular cost knapsack problem both have constant factor guarantees.

# 5 Experiments

In this section, we empirically compare the performance of the various algorithms discussed in this paper. We are motivated by the speech data subset selection application [20, 23] with the submodular function $f$ encouraging limited vocabulary while $g$ tries to achieve acoustic variability. A natural choice of the function $f$ is a function of the form $|\Gamma(X)|$, where $\Gamma(X)$ is the neighborhood function on a bipartite graph constructed between the utterances and the words [23]. For the coverage function $g$, we use two types of coverage: one is a facility location function $g_1(X) = \sum_{i \in V} \max_{j \in X} s_{ij}$ while the other is a saturated sum function $g_2(X) = \sum_{i \in V} \min\{\sum_{j \in X} s_{ij}, \alpha \sum_{j \in V} s_{ij}\}$. Both these functions are defined in terms of a similarity matrix $\mathbf{S} = \{s_{ij}\}_{i,j \in V}$, which we define on the TIMIT corpus [5], using the string kernel metric [27] for similarity. Since some of our algorithms, like the Ellipsoidal Approximations, are computationally intensive, we restrict ourselves to 50 utterances.

We compare our different algorithms on Problems 1 and 2 with $f$ being the bipartite neighborhood and $g$ being the facility location and saturated sum respectively. Furthermore, in our experiments, we observe that the neighborhood function $f$ has a curvature $\kappa_f = 1$. Thus, it suffices to use the simpler versions of algorithm EA (i.e., algorithm EASSCc and

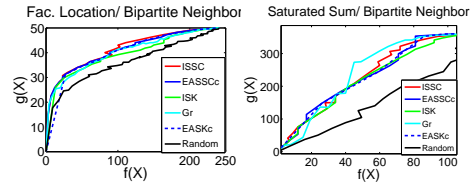

Figure 1: Comparison of the algorithms in the text.

EASKc). The results are shown in Figure 1. We observe that on the real-world instances, all our algorithms perform almost comparably. This implies, moreover, that the iterative variants, viz. Gr, ISSC and ISK, perform comparably to the more complicated EA-based ones, although EASSC and EASK have better theoretical guarantees. We also compare against a baseline of selecting random sets (of varying cardinality), and we see that our algorithms all perform much better. In terms of the running time, computing the Ellipsoidal Approximation for $|\Gamma(X)|$ with $|V| = 50$ takes about 5 hours while all the iterative variants (i.e., Gr, ISSC and ISK) take less than a second. This difference is much more prominent on larger instances (for example $|V| = 500$).

# 6 Discussions

In this paper, we propose a unifying framework for problems 1 and 2 based on suitable surrogate functions. We provide a number of iterative algorithms which are very practical and scalable (like Gr, ISK and ISSC), and also algorithms like EASSC and EASK, which though more intensive, obtain tight approximation bounds. Finally, we empirically compare our algorithms, and show that the iterative algorithms compete empirically with the more complicated and theoretically better approximation algorithms. For future work, we would like to empirically evaluate our algorithms on many of the real world problems described above, particularly the limited vocabulary data subset selection application for speech corpora, and the machine translation application.

**Acknowledgments:** Special thanks to Kai Wei and Stefanie Jegelka for discussions, to Bethany Herwaldt for going through an early draft of this manuscript and to the anonymous reviewers for useful reviews. This material is based upon work supported by the National Science Foundation under Grant No. (IIS-1162606), a Google and a Microsoft award, and by the Intel Science and Technology Center for Pervasive Computing.

## Footnotes

[1] A monotone non-decreasing normalized $(f(\emptyset) = 0)$ submodular function is called a polymatroid function.

[2]We can assume, w.l.o.g that $f(j) > 0, g(j) > 0, \forall j \in V$

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
