[Reviews · NeurIPS 2013]

Submitted by Assigned_Reviewer_4

I will refer only to extended version (supplement).

The paper considers two related discrete optimization problems:

min f(X) s.t. g(X)>=c

max g(X) s.t. f(X)<=b

where f and g are submodular set functions. These problems generalize
many problems from submdular optimization, which were treated inthe
literature before.


Contributions:

It is shown that the two problems are polytime reducible to each
other, even in approximation. For this reason, they are called
"dual" to each other.

A majorize-minimize (MM) meta-algorithm is proposed for both problems,
in which f and/or g are replaced by simpler surrogate functions, which
are majorants/minorants of the original functions tight at current
optimum. This is analogical to well-known difference-of-submodular
minimization problem and algorithm and to DC programming.

In particular, two types of these surrogates are considered: modular
functions and ellipsoidal approximations of the polymatroid polyhedron
[9]. The bounds these provide on the original submodular function are
yet refined using the notion of curvature of submodular function.

For both types of surrogates, the approximation bounds provided by the
MM algorithm are rigorously derived, being comparable or better than
the bounds known for the existing specialized algorithm for special
cases of both problems.

Furthermore, hardness results for the algorithm is derived. Precisely,
it is proved that no polytime algorithm can achieve a better
approximation than a certain guarantee. This guarantee is only
slightly better than the actual guarantee derived for the MM
algorithms.

The proposed algorithms were tested experimentally on real machine
learning problems (mid-size, no large-scale) with non-modular f and g.


Comments:

The paper is very rigorous, cleanly written and technically on high
level. The math is elegant and very clear (given complexity of the
topic). I have no major objections.

An obvious objection is that the experiments could be more
extensive. Eg, add a comparison with existing specialized algorithms,
add large-scale instances like cosegmentation [30]. But
the theoretical results are valuable per se.

On the negative side, the results are not very surprising, thus I consider the paper incremental (though very well written).

Minor issues:

- eq. (5): replace R^n with R^V

- l. 220: this inequality holds for every X, right?

- Write clearly if Lemma 2.1 is your contribution or not.

- Perhaps, I'd suggest to move section 3 before introduction of
surrogate functions, because it does not depend on them.

- I'm not sure whether it is right to call problems 1 and 2 "dual". It
seems to me there is more to duality than just polytime
reducibility, but you show just this. It seems to me that in
addition, a solution of one problem should be "readily" obtainable
from a solution of the other.

- l. 729: replace |X|\cap R| with |X\cap R|
Summary: Gives a novel unified view on a wide class of combinatorial optimization problems involving submodular function. The paper is clear and rigorous.

Submitted by Assigned_Reviewer_5

The authors introduce two new submodular optimization problems and
investigate approximation algorithms for them. The problems are
natural generalizations of many previous problems: there is a covering
problem (min f(X) st. g(X) >= c) and a packing or knapsack problem
(max g(X) st. f(X) <= b), where both f and g are submodular. These
generalize well-known previously studied versions of the problems
usually assume that f is modular. They show that there is an intimate
relationship between the two problems: any polynomial-time
bi-criterion algorithm for one problem implies one for the other
problem (with similar approximation factors) using a simple reduction.
They then present a general iterative framework for solving the two
problems by replacing either f or g by tight upper or lower bounds
(often modular) at each iteration. These tend to reduce the problem
at each iteration to a simpler subproblem for which there are existing
algorithms with approximation guarantees. In many cases, they are able
to translate these into approximation guarantees for the more general
problem. Their approximation bounds are curvature-dependent and
highlight the importance of this quantity on the difficulty of the
problem. The authors also present a hardness result that matches their
best approximation guarantees up to log factors, show that a number of
existing approximation algorithms (e.g. greedy ones) for the simpler
problem variants can be recovered from their framework by using
specific modular bounds, and show experimentally that the simpler
algorithm variants may perform as well as the ones with better
approximation guarantees in practice.


Quality: This is an extremely high quality paper that makes numerous
contributions. I especially like the fact that it presents a unifying
framework from which many specific results (new and old) can be
derived. Thus, it contributes not only to the compendium of
approximation results for specific submodular optimization problems,
but also to the general way of thinking about the research area. The
insights relating approximability to curvature seem to be a separate
and interesting thread of contribution. It is a very complete piece
of work.

Clarity: The paper is also impeccably written. It packs in many results and is
thus very dense. But it is well organized and follows a very logical
outline that first presents the main ideas and then uses them to
instantiate a number of specific results. My one suggestion to the
authors is that, for the short version of the paper, perhaps you can
consider omitting some of the lesser results to include a bit more
discussion. For example, the extensions in Section 4.3 could easily
be relegated to the longer version. Two things that would add to the
paper are: (1) some interpretation of approximability results and (2)
a bit more explanation of the experiments.

For example, I had the following question: it seems that the
ellipsoidal-based algorithms (e.g. EASSC) give the strongest
approximation-guarantees. How should I think about the relationship
between the guarantee given by ISSC and EASSC? In practice is the
EASSC bound usually better?

Originality: while there has been a significant amount of work on
submodular optimization, and this paper touches on many specific
pieces of prior work, it seems to be quite original in the way it
relates all of the prior work within a unifying framework.

Significance: I am not an expert in this area, but I think it could
be highly significant. One question is how often the most general
version of the problem (submodular cost + submodular covering
constraint) really crops up in practice vs. the previously studied
special cases.


Minor comments:
* Line 157: "A simple modification of the iterative procedure,
however, guarantees improvement". This seems like a counterpoint
to some previous point, but I can't figure out which. Perhaps it
was orphaned.
* Typo in Corollary 4.10: \sqrt{n} \log --> \sqrt{n} \log n
* The experiment setup is a bit hard to follow. It would be nice to
be able to understand this without referring to prior work. Can
you elaborate a bit on the application? It seems like you are
trying to select utterances which correspond to a small set of
words, while trying to capture the most variety in the
utterances. Is this right? What is the relevance of this problem?
Summary: This paper generalizes previous work on submodular optimization by
introducing two simple-to-state new optimization problems and a clean
and general framework to (approximately) solve them. In doing so, it
makes numerous specific contributions. Overall, it is a very polished
and complete piece of work.

Submitted by Assigned_Reviewer_6

* Summary

This well-written manuscript presents an in-depth analysis of two optimization problems, the minimization of a submodular function subject to a submodular lower bound, and the maximization of a submodular function subject to a submodular upper bound. Both problems are motivated by a number of practical applications. Unlike existing work that addresses these problems by minimizing a difference between submodular functions, this manuscript studies algorithms that solve the constrained problems approximately. Central to this work is the idea of quantifying approximate optimality and approximate feasibility in a definition of bicriteria algorithms. This idea is used to establish a form of duality of the problems as well as a number of interesting hardness results. The concept of bicriteria algorithms is potentially useful beyond this work.

* Strengths

- A comprehensive analysis of two important optimization problems is carried out meticulously, with in-depth knowledge of related work and command of the methods used.
- The manuscript is very well-written and a pleasure to read

* Weaknesses

- I do not see any major problem with this manuscript.
- One issue that might be worth discussing is the fact that the main manuscript contains almost too much information for a conference paper. The authors have chosen what I consider an acceptable solution, i.e., to forward cite an extended version which they provide as supplementary material.
Summary: A comprehensive analysis of two important optimization problems is carried out meticulously, with in-depth knowledge of related work and command of the methods used. The manuscript is very well-written and a pleasure to read.

Submitted by Assigned_Reviewer_7

The paper studies sub-modular optimization with sub-modular constraint. The authors present the duality between two types of related problems and approximation algorithms.

The paper address a problem which is of interest in many fields, including some applications in machine learning. It provides new results that advance the study of the field. The paper is very dense and at some points hard to follow. For example, the presentation of ellipsoidal approximation and the kappa curve normalization is made too early to my taste and distract the reader. Another example for that is at line 206 where the authors refer to an extended discussion of the bisection method in the extended material but even in the extended material this is discussed only to a small extent, I suggest dropping this discussion if the authors do not find it important or extending it if it is of value.

I did not find any flaws in the theoretical analysis. Theorem 3.1 can be improved by using binary search. In a sense, there is an upper bound U = g(V) and lower bound L = min_j(g(j)) and the goal is to find the maximal n such that U(1-\epsilon)^n\geq L and U(1-\epsilon)^n has a certain property described in algorithm 2 (or 3). The current description uses linear search, however binary search can be used to perform the search which will lead to a bound on the number of calls of O(log(log(U/L))).

The technique used in Algorithm 1 resembles the concave-convex procedure of Yuille and Rangarajan and I wish the authors would have referred to this similarity.

In lines 136-137 there is some un-clarity in the description since i\in V but i is not necessarily an integer number, instead it could be just an abstract object. The authors assume it is an integer number or otherwise assume order on the objects without explicitly saying that.

The plot in figure 1 is too small and does not allow the reader to see the difference between the lines (or the lack of difference).
Summary: The authors study constrained sub-modular optimization and provide new results. The analysis is comprehensive but the writing is dense and hard to follow at times.
Author Feedback

Author rebuttal: Firstly, we would like to thank all reviewers for their time and valuable reviews. Below, we address each reviewer in turn.

Assigned_Reviewer_4: Our primary goal (and the contribution) of our paper is to provide a complete theoretical treatment to our newly introduced submodular optimization problems (SCSC and SCSK). The experiments are more of a "proof of concept" for now, and show good results. We do find it interesting, however, that SCSC and SCSK have a form of approximation guarantee (lines 99-104), while DS (Eq. 1) does not (lines 59-60). This is distinct from, say, convex maximization or DC programming, thus making our results (i.e., the set function case) perhaps quite surprising.

Also, to our knowledge, there often really aren't specialized algorithms for some applications -- e.g., data subset selection (used in our experiments) where we simultaneously encourage limited vocabulary and acoustic diversity; there is prior work that investigates each individually (see [23, 25, 16] in the extended version), but not jointly. The reason is that the joint problem becomes one of either SCSC or SCSK, which, as far as we know, are first introduced in this paper. In the cosegmentation case, however, we agree it would be interesting to compare against the alternate procedures given in [26] and to do it at a large scale. Future work, moreover, will fully investigate additional applications (since submission, we've identified an important one in privacy/security).


Minor issues: We will incorporate all suggestions into the final version, if accepted.
Below are some clarifications:
1) Line 220, it holds for all X \subseteq V.
2) Lemma 2.1 *is* our contribution.
3) We agree that we use "duality" a bit loosely. We really mean poly-time "approximation preserving transformations".

Assigned_Reviewer_5: We will incorporate your suggestions related to the writing in the final version. With respect to ISSC and EASSC, the iterative algorithm ISSC is extremely practical and scalable (though with a slightly worse guarantee), while EASSC, which has the tightest guarantee, is very slow in practice since it involves computing the ellipsoidal approximations. We discuss this a bit in lines 254 and in the experiments sections (lines 412 - 416, all in the main paper).

Minor issues:
1) Line 157: This line pertains to the fact that the improvement in the Algorithm, is guaranteed only if the subproblems can be exactly solved. In the case when they are only approximately solvable, we need to slightly modify the procedure - we have a detailed discussion of this in the extended version (lines 210-212). We will clarify
this in the final version, if accepted.
2) We will elaborate on the experimental setup and the specific application in the final version, if accepted. We do, however, have some discussion on the specific applications in lines 74 - 79 of the main paper.

Assigned_Reviewer_6: Thank you very much for your encouraging review!